# Epigenetic Modifications Associated with Maternal Anxiety during Pregnancy and Children’s Behavioral Measures

**DOI:** 10.3390/cells10092421

**Published:** 2021-09-14

**Authors:** Lei Cao-Lei, Marion I. van den Heuvel, Klaus Huse, Matthias Platzer, Guillaume Elgbeili, Marijke A. K. A. Braeken, Renée A. Otte, Otto W. Witte, Matthias Schwab, Bea R. H. Van den Bergh

**Affiliations:** 1Schizophrenia and Neurodevelopmental Disorders Research Section, Douglas Hospital Research Centre, Montreal, QC H4H 1R3, Canada; lei.cao@mail.mcgill.ca (L.C.-L.); Guillaume.Elgbeili@douglas.mcgill.ca (G.E.); 2Department of Cognitive Neuropsychology, Tilburg University, 5037 AB Tilburg, The Netherlands; m.i.vdnheuvel@tilburguniversity.edu; 3Genome Analysis Group, Leibniz Institute on Aging—Fritz Lipmann Institute, 07745 Jena, Germany; klaus.huse@t-online.de (K.H.); matthias.platzer@leibniz-fli.de (M.P.); 4Rehabilitation Research Center, Biomedical Research Institute, Faculty of Rehabilitation Sciences, Hasselt University, 3500 Hasselt, Belgium; marijke.braeken@uhasselt.be; 5Family Care, Department of Personal and Preventive Care, Philips Research, 5656 AE Eindhoven, The Netherlands; renee.otte@philips.com; 6Hans Berger Department of Neurology, Jena University Hospital, 07740 Jena, Germany; otto.witte@med.uni-jena.de (O.W.W.); matthias.schwab@med.uni-jena.de (M.S.); 7Health Psychology Research Group, University of Leuven (KU Leuven), 3000 Leuven, Belgium; 8Department of Welfare, Public Health and Family, Flemish Government, 1030 Brussels, Belgium

**Keywords:** DNA methylation, *NR3C1*, *IGF2/H19*, *LINE1*, maternal anxiety, interaction, mediation, sex-specific effects

## Abstract

Epigenetic changes are associated with altered behavior and neuropsychiatric disorders and they modify the trajectory of aging. Maternal anxiety during pregnancy is a common environmental challenge for the fetus, causing changes in DNA methylation. Here, we determined the mediating role of DNA methylation and the moderating role of offspring sex on the association between maternal anxiety and children’s behavioral measures. In 83 mother–child dyads, maternal anxiety was assessed in each trimester of pregnancy when the child was four years of age. Children’s behavioral measures and children’s buccal DNA methylation levels (*NR3C1*, *IGF2/H19 ICR*, and *LINE1*) were examined. Higher maternal anxiety during the third trimester was associated with more methylation levels of the *NR3C1*. Moderating effects of sex on the association between maternal anxiety and methylation were found for *IGF2/H19* and *LINE1* CpGs. Mediation analysis showed that methylation of *NR3C1* could buffer the effects of maternal anxiety on children’s behavioral measures, but this effect did not remain significant after controlling for covariates. In conclusion, our data support an association between maternal anxiety during pregnancy and DNA methylation. The results also underscore the importance of sex differences and timing effects. However, DNA methylation as underlying mechanism of the effect of maternal anxiety during pregnancy on offspring’s behavioral measures was not supported.

## 1. Introduction

With the population aging worldwide, understanding the biology of healthy aging is more relevant than ever. Brain aging is a major determinant of aging. Individual rates of brain aging—including age-related brain diseases such as neurocognitive and neuropsychiatric disorders—may be shaped not only by the genome but also by the epigenome [1,2,3], which is extensively modified by the prenatal environment [4]. Epigenetic modifications allow the offspring to rapidly adapt its phenotype in response to environmental cues at the expense of a predisposition for disease in order to reach the reproduction period [5]. The relationship between adverse environmental influences during critical periods of prenatal life and the health of offspring in later life is the basis of the ‘Fetal Programming’ or ‘Developmental Origins of Health and Disease (DOHaD) hypothesis’ [6].

Exposure to maternal distress during pregnancy—including the experience of anxiety, of depression or of stressful events—is a rather common environmental challenge for the fetus. About 30% to 40% of pregnant women experience psychosocial distress (broadly defined) during pregnancy [7,8,9,10]. With regard to anxiety, 18.2%, 19.1%, and 24.6% experience self-reported anxiety in the first, second, and third trimesters, respectively (meta-analysis; Dennis et al. [9]) and 14% to 15.8% is diagnosed with an anxiety disorder [11]. The prevalence of depression is up to 10% [12] or 7.4% (95% CIs: 2.2, 12.6), 12.8% (95% CIs: 10.7, 14.8), and 12.0% (95% CIs: 7.4, 16.7) for each trimester, respectively [13]. Differences in prevalence depend on population characteristics, timing, and type of screening used [11]. Higher percentages for depression (pooled prevalence of 25.5% in 37 studies (*n* = 47,677)) and anxiety (pooled prevalence of 30.5% in 34 studies (*n* = 42,773)) occur during periods of universal stress exposure, such as the corona pandemic [14]. Boekhorst et al. [15] reported a 49.7% higher distress score in the COVID pandemic group as compared to a pre-pandemic group.

Exposure to maternal distress may increase fetal stress sensitivity and adapt fetal brain function to meet the challenges of an “expected” stressful postnatal environment [5,16,17,18]. However, a stress-sensitive brain is susceptible to stress-related disorders—such as behavioral measures, mental health problems, and neuropsychiatric diseases—in later life in a sex-specific manner [19,20]. Although the aging brain is particularly vulnerable due to its loss of resilience [21], neurodevelopmental and behavioral problems become phenotypically apparent already at childhood [22,23,24,25,26]. This is important for human research since prospective studies from pregnancy to old age are extremely time consuming.

Still, it remains unclear how maternal distress during pregnancy “gets under the skin” of the offspring. Epigenetic mechanisms are well known to be involved in the individual trajectory of brain development and programming of the activity of the stress axis [27,28,29,30,31]. For instance, DNA methylation of the glucocorticoid receptor gene (*NR3C1*) may increase offspring stress sensitivity. In one seminal paper from human study, Oberlander’s group revealed that prenatal maternal anxiety and mood disorders were associated with increased *NR3C1* exon 1F DNA methylation in leukocytes from cord blood, which was also associated with an increased salivary cortisol response in three-month-old infants [32]. Other promoter regions of *NR3C1* have also drawn attention. For instance, the methylation profile of *NR3C1*, *1B*, *1D*, and *1F* promoter regions was investigated in cord blood mononuclear cells trigged by the effect of maternal distress during pregnancy [33]. More recently, the DNA methylation within proximal (within and at the shores of the CpG island) and distal promoter regions was investigated in a rat study [34], showing tissue-, sex-, and age-specific DNA methylation of these regions. Evidence from human studies also showed sex-specific DNA methylation in offspring of mothers experiencing stress during pregnancy [35,36]. Although a meta-analysis [30] and several review papers [4,37,38] reported mixed results (i.e., no changes, decreased and increased methylation), in the aftermath of prenatal exposure to objective hardship (war-related trauma, interpersonal violence) studies focusing on self-reported maternal distress in pregnancy mostly report positive correlations with *NR3C1* methylation (however, Mansell et al. [39] reported no effect). The authors argued that these mixed findings were largely due to the lack of methodological consensus, cell- or tissue specific effects (either neonatal cord blood, or whole blood, or placental tissue, or buccal cells were analyzed), and selection of CpG sites. Additionally, they noted an extreme focus on the *1F* exon and emphasized the need for widening the examined sequence, in order to include all non-protein-coding first exons of the *NR3C1* in the analysis [30]. Taken together, while several experimental and human studies relate maternal objective hardship as well as maternal distress in pregnancy to methylation changes of *NR3C1*, there is some evidence of an association of such methylation changes with offspring psychosocial stress reactivity (e.g., [32,40]) and behavior (e.g., [41]) and hardly any evidence for the role of methylation changes of *NR3C1* in the association between maternal distress in pregnancy and offspring behavior [35] (for a review, see Cao Lei et al. [4]; Berretta et al. [37]; Sosnowski et al. [42]).

Maternal mental health during pregnancy has not only been associated with DNA methylation of the *NR3C1* in the offspring, but also with other genes such as imprinted genes. For instance, the *IGF2* encodes insulin-like growth factor 2 which is an important growth hormone for fetal development [43]. The *IGF2* consists of several differentially methylated regions (DMRs) located throughout *IGF2* and the neighboring *H19*. The imprinted control region (*ICR*) located between *IGF2* and the upstream *H19* is involved in expression of *IGF2* paternally inherited allele [44]. Chen et al. [45] found increased DNA methylation of *IGF2/H19 ICR* in placental and cord blood of children whose mothers experienced high levels of distress. However, in another study, maternal anxiety was associated with decreased *IGF2/H19 ICR* DNA methylation in cord blood of female neonates [39]. These inconsistent findings need to be studied further. Moreover, only some human studies relating methylation changes of *IGF2/H19 ICR* to behavioral outcome are available [46,47,48]. A possible explanation for the inconsistent findings could be that early life stress alters epigenetic patterns in a sex-specific manner, potentially under the control of sex chromosomes and/or sex hormones [49,50,51]. Yet, sex differences are often ignored in research on epigenetic effects of early life stress [52]. Similarly, timing effect of exposure during pregnancy may constitute an important factor for outcome in later life [53], and data from the first trimester of pregnancy is often lacking (e.g., [32]). Therefore, sex effect and timing effect need to be taken into account as critical factors moderating the effect of prenatal exposure to maternal distress on the fetal epigenome.

The current study prospectively investigated whether maternal anxiety during pregnancy could influence children’s behavioral measures through epigenetic mechanisms. We considered the timing of exposure to maternal anxiety on child’s DNA methylation status, which seems to have significant effects on offspring behavior (reviewed in Van den Bergh et al. [19]). Therefore, the aims of this study were to: (1) determine the association of maternal anxiety in first, second, and third trimester with buccal cell DNA methylation of candidate regions in four-year-olds, and, the moderating role of sex on this association; (2) determine the mediating role of buccal DNA methylation on the association of maternal anxiety during pregnancy with children’s behavioral measures and the moderating role of sex on the mediation effect. We selected three candidate regions for our investigation: *NR3C1*, *IGF2/H19 ICR*, and long interspersed nucleotide elements 1 (*LINE1*). *NR3C1* and *IGF2/H19* are prime targets in mediating effects of prenatal stress and offspring neurodevelopment and behavior [34]. *LINE1* are quantified as an indicator of global methylation status [54].

## 2. Materials and Methods

### 2.1. Study Design and Participants

The authors assert that all procedures contributing to this work comply with the ethical standards set by the St. Elisabeth hospital Ethical Review Committee on research regarding human subjects and with the Helsinki Declaration. All participating parents provided written informed consent.

Data were collected as part of the Prenatal Early Life Stress (PELS) project, an ongoing prospective cohort study in Tilburg, The Netherlands, following pregnant women and their offspring from the beginning of pregnancy onwards. Participants were recruited before the 15th week (*n* = 178; between the 8th and 14th week) and between the 15th and 22nd week of pregnancy (*n* = 12) from the St. Elisabeth hospital in Tilburg, The Netherlands, and four midwife practices. A total number of 191 children were born in this study (one pair of twins). Via postal or digital questionnaires mothers provided information on their psychological state three times during pregnancy (once every trimester), and three times after birth (at 2/4 months, at 9 months, and at 4 years). At four years of age, buccal cells were collected from the children. For the purpose of the current study, we analyzed data of all mother–child dyads with available questionnaires of maternal prenatal anxiety and child buccal cells.

From a total of 118 children (one twin) buccal cells were collected. Reasons for not participating in the buccal cells collection were: decline or no response to the invitation for participation (*n* = 35), loss to follow-up (*n* = 20), drop-out (*n* = 13; i.e., families that dropped-out at earlier waves), or miscarriage/death/disability of the child (*n* = 5). Part of the final sample was excluded (*n* = 9) because of too little DNA and/or the quality of the DNA being too low. In addition, we excluded all mother–child dyads of which the mother smoked during pregnancy or did not disclose whether they smoked or not (*n* = 15) and of which the children had been born prematurely or with a low birth weight (*n* = 5) (gestational age ≤ 36 and/or birth weight ≤ 2600 g). Finally, those mother–child dyads that did not have maternal prenatal anxiety data were excluded. The final sample differed per trimester, since not all mothers completed the questionnaires at every wave: *n* = 82 for the first trimester, *n* = 83 for the second trimester, *n* = 83 for the third trimester.

Our final sample consisted of 39 boys and 44 girls. Almost all participating mothers were Caucasian, except for on mother who reported Asian as her ethnical background. The nationality of our mothers was mostly Dutch, with some mothers reporting double nationalities (i.e., Russian, Romanian, French, and German).

### 2.2. Questionnaires Measuring Maternal Anxiety and Children’s Behavioral Measures

Maternal anxiety. Maternal anxiety during pregnancy was assessed with the Symptom Checklist-90 (SCL-90) [55]. The anxiety subscale of the SCL-90 mainly measures somatic anxiety symptoms (e.g., vegetative arousal) instead of merely psychological anxiety symptoms (e.g., anxious thoughts). Participants rated the scale, which consists of 10 items, on a five-point Likert scale (1 = not at all, 2 = somewhat, 3 = quite, 4 = quite a lot, and 5 = extremely). A higher score indicates a higher level of experienced anxiety. In general, the scale has good convergent and divergent validity and has good internal consistency (α = 0.88) [55]. In our sample, the scale had high reliability during pregnancy (trimester 1: α = 0.876; trimester 2: α = 0.828; trimester 3: α = 0.803). The same questionnaire was used again for the measurement of maternal anxiety when the child was four years of age and had a Cronbach’s alpha of α = 0.90 at this measurement point.

Child behavioral measures. When the children were four years of age, mothers reported on potential behavioral measures of their children using the Child Behavioral Checklist (CBCL) [56]. The CBCL consists of 99 items about the children’s behavior with a three-point Likert scale ranging from ‘not at all’ (0) to ‘clearly’ or ‘often’ (2). For the purpose of this study, we used the internalizing and externalizing subscales and the Diagnostic and Statistical Manual of Mental Disorders (DSM)-IV specific subscales, including affective symptoms, anxiety symptoms, pervasive developmental symptoms, attention deficit/hyperactivity symptoms, and oppositional defiant symptoms. In our sample, four children had clinical levels of affective problems, seven had clinical levels of anxiety problems, three had clinical levels of pervasive developmental disorder, three had clinical levels of Attention-Deficit/Hyperactivity Disorder (ADHD) problems, and four had clinical levels of oppositional defiant problems.

### 2.3. Covariates

Based on previous research on the association between maternal psychological functioning during pregnancy and DNA methylation status [33], we considered a number of possible covariates, including: birth weight of the child, gestational age at birth, child gender, and SCL-90 maternal anxiety when the child was four years of age.

### 2.4. Buccal Cells Collection, DNA Extraction, and Bisulfite Treatment

We used a standardized protocol for buccal cell collection from the children. In short, children were told we were going to “brush their cheeks on the inside for 10 seconds”. We used a Cytosoft^TM^ cytology brush (Thermo Fisher Scientific, Waltham, MA, USA) and we rotated it five times on one side and five times on the opposite interior cheek. The head of the brush was placed in a tube with buffer which contains Proteinase K. DNA extraction was performed using PrepIT-L2P kits (DNA Genotek Inc., Ottawa, ON, Canada) according to the manufacturer’s instructions. The average yield was around 2 µg. DNA was stored at −20 °C until analysis. Bisulfite-converted DNA was subjected to PCR amplification of the CpG regions under study.

### 2.5. Bisulfite (BS) Specific PCR

#### 2.5.1. *NR3C1*

Two amplicons were selected from the human *NR3C1* promoter regions: four CpGs named CpG1–CpG4 (283bp; Chr5: 143434589–143434871) which are located in exon *1A*, and six CpGs named CpG5–CpG10 (178bp; Chr5: 143401889-143402066) located in the intron between exon *1H* and exon2 (Figure 1A). The genomic position of each CpG is shown in Appendix A. Primers were designed using MethPrimer (https://www.urogene.org/methprimer/). A nested PCR approach was applied. The first PCR was performed with the same pairs of amplicon-specific BS primers for all samples in microtiter plates. Cycling was done using the Bioline PCR mix (BIOLINE, Luckenwalde, Germany) in a 25 µL reaction vol. (1 min denaturation at 94 °C, followed by 29 cycles at 57 °C for 30 s, 72 °C for 1 min, 94 °C for 30 s, and a final elongation step at 72 °C for 5 min. PCR products was diluted 1:10 and 1 µL of the dilution was used in a second PCR using sample-specifically tagged primers (tag: 3 nucleotides at 5 = -ends; Appendix A). PCR conditions were the same as in the first PCR, however, number of cycles was reduced to 25. Successful amplification was checked by electrophoresis on 1% agarose gels.

#### 2.5.2. *IGF2/H19* Imprinting Control Region (*IGF2/H19 ICR*)

One region including 10 CpGs (123bp; Chr11: 1999847–1999969) from the human *IGF2/H19 ICR* was amplified (Figure 1B). The genomic position of each CpG is shown in Appendix A. Primers are shown in Appendix A. PCR conditions are as for *NR3C1* amplification.

#### 2.5.3. *LINE1* Motifs

*LINEs* are retrotranposons with thousands of copies within the human genomes. A significant portion of global DNA methylation is found in these loci and *LINE* methylation may be considered a proxy for global methylation [54]. Amplification of many *LINE1* copies in parallel was performed with primers designed using a consensus sequence of most prominent *LINE1* families. Primers were taken from Gries et al. [57] (forward primer: TTATTAGGGAGTGTTAGATAGTGGG, reverse primer: CCTCTAAACCAAATATAAAATATAATCT) and amplification performed as above.

### 2.6. Sequencing and Data Processing

In general, we applied a multiplex next-generation deep sequencing approach called “BS Amplicon Sequencing” [34] combining the high sensitivity of BS pyrosequencing in respect to the analyses of single or tight neighboring CpGs with the advantage of BS Sanger sequencing, providing joint methylation information of all CpGs within PCR amplicons. Moreover, this approach is highly cost and labor effective. Sequencing, demultiplexing, data extraction, and methylation analysis for the single copy loci *NR3C1* and *IGF2/H19 ICR* were performed as previously described in Agba et al. [34]. In short, sequences of reads with correct primer and tag ends and expected size are aligned and CpG positions inspected for harboring C (methylated) or T (unmethylated). Methylation was inferred as C/C+T.

For the diverse *LINE1* reads, alignment across the entire read length is error prone. Therefore, we determined *LINE1* methylation by inspecting each individual read sequence for harboring a CpG, and for each of the identified CpGs a motif of the structure N10(CpG)N10 was inferred (N stands for A, C, G, or T). To quantify methylation, all N10(CpG)N10 and corresponding N10(TpG)N10 motifs were counted in the entire dataset (TpG is regarded as a signature for non-methylated CpG after BS conversion and amplification). Three motifs were identified as the most common ones in all LINE1 reads. Therefore, they (motif1: AGATAGTGGGYGTAGGTTAGTG, motif2: TTTGGAAAATYGGGTTATTTTT, motif3: ATTTGGGAAGYGTAAGGGGTTA) were selected for LINE1 methylation evaluation. Software tools were developed for the filtering of sequences according to primer and tag integrity as well as for size (‘bucketer’), and for the extraction of sequence motifs (‘sad’). Respective scripts (C++) can be downloaded from http://genome.leibniz-fli.de/software/buck_sad/buck_sad.tgz.

All methylation rates are corrected for deamination efficiency, which was determined by inspecting three putatively unmethylated CT positions in the *NR3C1* and *IGF2/H19 ICR* amplicons.

### 2.7. Statistical Analyses

Correlation analyses were performed between the covariates (child: weight and gestational age at birth, gender; mother: SCL-90 anxiety when the child is four years of age), DNA methylation levels, and child behavioral problems. Pearson correlation was performed.

Next, linear regression analyses were conducted to determine whether maternal anxiety at trimesters 1, 2, and 3 had an effect on DNA methylation, and whether child’s sex moderated the effect of maternal anxiety on child’s DNA methylation level.

To investigate whether the three genes mediated the relationship between exposure to maternal anxiety and children’s behavior outcomes, we conducted mediation analyses using bootstrapping, conducted with the SPSS procedure PROCESS macro [58]. The mediator model was modeled using multiple regression models with children’s behavior as the outcome; maternal anxiety at trimesters 1, 2, and 3 as predictors; and DNA methylation levels of CpGs as mediators. The model of the mediation analysis is presented in Figure 2. Path “a” is the effect of the predictor variable on the DNA methylation (mediator), path “b” is the effect of the DNA methylation on the outcome variable controlling for the predictor variable, path “c’ ” is the direct effect of the predictor variable on the outcome variable controlling for DNA methylation (mediator). The coefficient “a × b” represents the “indirect” (or mediating) effect of the predictor variable on the outcome variable through DNA methylation (mediator). Path c = c’ + ab, represents the total effect and is derived by summing the direct and indirect effect. We tested the indirect effects (mediation effect) of prenatal maternal anxiety on the child behavioral measures through each CpG site by computing 95% bias-corrected bootstrap confidence intervals, in accordance with Hayes [59]. The SPSS procedure PROCESS macro was used to conduct the analyses. Each bootstrap resampled the initial sample 10,000 times. A mediation effect was considered significant if 0 was not included in the bootstrap confidence interval.

All statistical analyses were performed using IBM (https://www.ibm.com) SPSS version 22.0 for Windows using α = 0.05.

## 3. Results

### 3.1. Participants’ Characteristics; Descriptive Analysis

Descriptive statistics for gestational age at birth, maternal anxiety levels during the three pregnancy trimesters and when child was four years of age, children’s birth outcomes, and behavior outcomes at four years of age are presented in Table 1. Descriptive statistics for children’s characteristics in boys and girls separately are presented in Appendix A. There were no significant differences between boys and girls for our predictor variables or confounders. Descriptive statistics for DNA methylation levels are presented in Appendix A.

Correlation of children’s behavioral measures with maternal anxiety during pregnancy and with DNA methylation levels are shown in Table 2. Maternal anxiety during the second trimester was significantly associated with children’s internalizing problems (*p* = 0.025) and DSM oppositional defiant problems (*p* = 0.03). Maternal anxiety during the third trimester was significantly associated with children’s DSM attention deficit/hyperactivity problems (*p* = 0.026). Neither maternal anxiety during the first trimester nor methylation of any CpG site was significantly associated with any children’s behavioral measures.

Gestational age at birth was significantly (*p* < 0.05) associated with maternal anxiety during the second trimester, DNA methylation of *NR3C1*-CpG8, and DSM attention deficit/hyperactivity problems. Maternal anxiety at four years of age was significantly (*p* < 0.05) associated with maternal anxiety during pregnancy (first, second, and third trimester) and children’s behavioral outcomes (see Appendix A). Therefore, gestational age at birth and maternal anxiety at four years of age were controlled for in each model.

### 3.2. Association of Maternal Anxiety with Child’s Buccal DNA Methylation Level

Maternal anxiety during the third trimester was significantly associated with the methylation level of four CpGs from *NR3C1* after controlling for gestational age at birth and maternal anxiety when the child was four years of age: *NR3C1*-CpG3 (β = 0.272, *p* = 0.020), CpG5 (*β* = 0.338, *p* = 0.005), CpG6 (β = 0.271, *p* = 0.023), and CpG10 (β = 0.345, *p* = 0.004). Higher maternal anxiety predicted higher *NR3C1* methylation level, explaining 6.7% of the variance of methylation level for *NR3C1*-CpG3, 10.3% for *NR3C1*-CpG5, 6.6% for *NR3C1*-CpG6, and 10.7% for *NR3C1*-CpG10. There was no effect of maternal anxiety in the third trimester on other CpGs of *NR3C1*, or of anxiety in the first and second trimester on any of the CpG of *NR3C1*.

For the 10 CpGs from *IGF2/H19 ICR* and *LINE1* motifs 1, 2, and 3, no significant association was found between maternal anxiety (first, second, and third trimester) and methylation levels.

### 3.3. Interaction Effect of Sex on the Association between Maternal Anxiety and Child’s Buccal DNA Methylation Level

Trimmed results of the linear regressions are presented in Table 3, Table 4 and Table 5. Sex moderated the effect of maternal anxiety during third trimester on DNA methylation levels; i.e., we observed significant interaction effects of sex on the association between maternal anxiety during the third trimester and DNA methylation level of *IGF2/H19*-CpG1, 4, and 6, explaining 5.4% (Table 3), 6.1% (Table 4), and 5.9% (Table 5) of the variance, respectively. As shown in Figure 3A–C, for boys, there was a negative association between maternal anxiety during the third trimester and the methylation level of *IGF2/H19* CpGs, with higher maternal anxiety predicting lower methylation levels. However, for girls, there was a positive association between maternal anxiety during the third trimester and the methylation level of *IGF2/H19* CpGs, with higher maternal anxiety predicting higher methylation levels. There were no sex interactions in the third trimester on other CpGs of *IGF2/H19*, or on the effect of maternal anxiety in the first and second trimester on any of the CpG of *IGF2/H19*.

Additionally, a significant sex × maternal anxiety during the second trimester-interaction effect on *LINE1* motif2 DNA methylation was found, explaining 9.9% of the variance (Table 6). As shown in Figure 3D, higher maternal anxiety in the second trimester was associated with greater methylation in boys, but lower methylation in girls. This interaction effect was only significant at maternal anxiety scores of 13.42 or higher; at lower levels, there were no sex differences observed. Furthermore, there was no interaction effect of maternal anxiety with sex in the second trimester on other CpGs of *LINE1* motif 1, or of anxiety in the first and third trimester on any of the CpG of *LINE1* motifs. There were no significant sex interactions for DNA methylation of any *NR3C1* CpGs.

### 3.4. Mediation Effect of DNA Methylation Level on the Association between Prenatal Maternal Anxiety and Child Behavioral Measures

As maternal anxiety during the third trimester was positively associated with the methylation level of *NR3C1*-CpG3, 5, 6, and 1, we further tested whether the methylation level of these CpGs could mediate the effect of maternal anxiety on children’s behavioral problems.

From the mediation model (Figure 4A), we observed that the methylation level of *NR3C1*-CpG10 negatively mediated the effects of maternal anxiety during the third trimester on children’s anxiety (a × b = −0.0462). The higher maternal anxiety during the third trimester, the greater the methylation of the *NR3C1*-CpG1, which in turn was then associated with lower anxiety symptoms in the children. In other words, although the direct effect (c’) of maternal anxiety on the children’s anxiety symptoms was positive (higher maternal anxiety, higher children’s anxiety), the mediation via DNA methylation of *NR3C1*-CpG10 seems to buffer the effect of maternal anxiety. The confidence interval of the mediation was between −0.1306 and −0.0006 in which 0 was not included, i.e., the mediation effect was significant (Appendix A). However, when controlling for the covariates gestational age at birth and maternal anxiety when the child was four years of age, the mediation effect of DNA methylation of *NR3C1*-CpG10 disappeared (*p* = 0.1796).

Similarly, the methylation level of *NR3C1*-CpG3 negatively mediated the effects of maternal anxiety during the third trimester on children’s ADHD (a × b = −0.0360) (Figure 4B). The higher maternal anxiety during the third trimester, the higher the methylation of the *NR3C1*-CpG3, which was subsequently, associated with lower children’s ADHD traits. The confidence interval was between −0.1190 and −0.0019 in which 0 was not included, i.e., the mediation effect was significant (Appendix A). Again, when controlling for the covariates gestational age at birth and maternal anxiety when the child was four years of age, the significant effect disappeared (*p* = 0.1812). No significant mediation effect was found for *NR3C1*-CpG5 and CpG6.

Next, we ran moderated-mediation analyses to explore whether the significant mediation effects were different between boys and girls (cf. theoretical model in Figure 2). None of the moderated-mediation models with sex as a moderator were significant, indicating that the mediation effects were not dependent on sex.

### 3.5. Sensitivity Analyses

To check whether our results were driven by a few extreme cases, we performed outlier analyses in SPSS for scores on maternal anxiety during pregnancy and child behavioral problems. Three extreme cases were detected for maternal anxiety and one extreme case for child behavioral problems (*n* = 4 total outliers). Analyses were rerun without the outliers and yielded similar results.

## 4. Discussion

The main goal of this study was to test whether maternal anxiety during pregnancy could influence children’s behavioral measures at four years of age through epigenetic mechanisms, and to elucidate the potential role of children’s sex. We proposed a model in which methylation levels of selected candidate regions mediate the association between maternal anxiety and child behavioral measures and in which this mediation could be dependent on sex of the child. We observed that children exposed to higher maternal anxiety during the third trimester had higher methylation levels of four CpGs from *NR3C1* (i.e., CpG3, CpG5, CpG6, CpG10). We also observed that maternal anxiety during pregnancy had an effect on the methylation levels of *IGF2/H19*-CpG1, 4, and 6 and *LINE1* motif2, with different directions of the effect for boys and girls. While we observed lower DNA methylation in *IGF2/H19*-CpG1, 4, and 6 in boys exposed to higher levels of maternal anxiety in the third trimester, we observed higher DNA methylation in these locations for girls. For *LINE1* motif2, we observed opposite findings (higher methylation in boys, lower methylation in girls). Furthermore, the methylation level of *NR3C1*-CpG10 and -CpG3 negatively mediated the effects of maternal anxiety during the third trimester on children’s behavioral problems. However, the mediation effect disappeared when controlling for gestational age at birth and maternal anxiety when child was at four years of age.

The positive association between maternal anxiety and methylation of CpGs from *N3RC1* corroborates the findings of studies that reported increased methylation of *NR3C1* in offspring prenatally exposed to maternal distress [32,33,36,40,60]. In some studies, it was demonstrated that maternal distress early in gestation has effects on cognitive and behavioral measures in the offspring, while other studies showed effects of maternal distress in late pregnancy (reviewed in Van den Bergh et al. [19]). In the current study, effects of maternal anxiety on DNA methylation level of *NR3C1* were only observed in the third trimester. This timing dependent relation is in line with the finding from Oberlander et al. [32], who were the first to study the association between maternal depressive symptoms during pregnancy and the methylation level of *NR3C1* in cord blood in newborns. They also reported a positive association between methylation of *NR3C1* and maternal depressed mood in the third trimester, but in other CpGs than in our study. These and our results indicate that maternal mental health during pregnancy has the strongest effects on *NR3C1* methylation at the end of pregnancy.

In animal studies, it was reported that vulnerability of the offspring to the influence of prenatal exposure to stress is moderated by offspring sex [61,62,63]. In human studies, it has been shown that prenatal factors could affect gene-specific epigenetic changes in offspring—such as in *IGF2/H19* [39], *HSD11B2* [64,65], and exon *1F* of *NR3C1* [35,36]—in a sex-specific manner. In the current study, we observed a significant interaction effect between maternal anxiety and offspring sex on the methylation level of the CpGs of *IGF2/H19 ICR* and *LINE1* motif2 but not of the CpGs of *NR3C1*. For *IGF2/H19 ICR*, we found a negative association between maternal anxiety and methylation levels in boys (Figure 3A–C), in contrast to a positive association in girls. Thus, we added evidence to the suggestion that maternal anxiety during pregnancy can influence *IGF2/H19* in a sex-specific manner [39]. The IGF2 serves different biological functions—such as regulation of cell proliferation, growth, migration, differentiation, and survival—and is differentially expressed in different tissues and at different developmental periods. Although we did not examine the specific effects of *IGF2* DNA methylation, our data suggest that, in girls exposed in utero during third trimester, methylation of *IGF2*—which could further lead to decreased *IGF2* expression—may be involved in offspring neurodevelopment shaping behavioral measures. However, the interaction effect of sex only explained a small proportion of the variance (5.4%), and more research is necessary before a firm conclusion can be drawn. Interestingly, maternal anxiety in the second trimester was found to interact with sex on the methylation level of *LINE1* motif 2, suggesting a vulnerability of the epigenome to maternal anxiety not only at the end of pregnancy which could also be sex-dependent and timing-dependent. Serving as surrogate marker for global DNA methylation, *LINE1* is the most abundant family of non-long terminal repeat retrotransposons in the human genome, accounting for around 17% of the genome [66]. In order to inhibit the expression of these repetitive sequences, the CpGs are normally highly methylated. They are also transposable, which means that their production might lead to their insertion into other genomic areas, effectively silencing genes. For instance, DNA methylation of *LINE1* was reported to be associated with biomarkers of metabolic health [67]. Moreover, Kile et al. [68] reported positive correlations between the DNA methylation of *LINE1* in maternal blood and that in umbilical cord blood of her child. The failed proof of a moderating role through offspring sex on the association between maternal anxiety and child behavior contradicts the study of Oberlander and colleagues [32]; however, this study examined other CpGs of *NR3C1* and only revealed a trend-level association between maternal depression and increased *NR3C1* methylation for female, but not male, infants.

Most studies on epigenetic regulation of *NR3C1* expression have investigated the promoter region upstream of rodent exon *1_7_* and its human ortholog *1F*. In this study, we focused on two other regions of *NR3C1*: exon *1A* and the 3’ CpG island shore between exon *1H* and exon 2. We examined how CpGs from *NR3C1* could mediate the effect of maternal anxiety on children’s outcomes. Without controlling for the covariates (gestational age at birth and maternal anxiety when child was at four years of age) CpG10 demonstrated a significant negative mediation effect on the association between maternal anxiety (third trimester) and children’s anxiety, and CpG3 demonstrated a significant negative mediation effect on the association between maternal anxiety (third trimester) and children’s ADHD at four years of age. Based on these results, we suggest that the effect of maternal anxiety (third trimester) on children’s anxiety could be buffered by DNA methylation of the two *NR3C1* regions. Such a ‘protective’ role of DNA methylation was also observed in previous studies [69,70]. Furthermore, a similar epigenetic mechanism was reported by a group of researchers [71], showing that maternal adversities during pregnancy predicted increased DNA methylation of oxytocin receptor gene in cord blood and suggesting that activity of oxytocin receptor expression could provide a mechanism by which the newborn adapts to a potentially challenging environment. Therefore, we could hypothesize that the methylation of this *NR3C1*-CpG10 and -CpG3 both change *NR3C1* gene expression, and further protect the children from an at-risk behavior outcome, rather than explaining how maternal anxiety predicts increased risk of behavioral problems in children via DNA methylation. However, after controlling for the covariates gestational age at birth and maternal anxiety, the significant mediation effects of prenatal maternal anxiety on children behavioral measures disappeared. This suggests that both gestational age at birth and concurrent maternal anxiety are important factors, which influences the epigenetic level of buccal cells of children at four years of age. Therefore, concurrent maternal anxiety should be considered in further studies [37].

Several limitations of our study should be mentioned. Although we showed changes in the DNA methylation status of the selected regions, it remains unclear whether this influences the expression of the corresponding genes as their expression levels were not assessed. In other words, based on this data alone, we cannot evaluate whether the DNA methylation changes observed are meaningful in altering child behavior directly. In addition, as the saliva samples and child behavior questionnaires were taken at the same time when the child was four years old, our results do not allow any definitive conclusions about the direction of cause between DNA methylation and behavioral problems. Besides, the sample size is limited; this may increase the chances of errors. Finally, our cohort is relatively high functioning and includes only a few cases with clinical levels of child behavioral problems. However, our study has major strengths as well: it is prospective, includes all three trimesters, explores several candidate genes, and it is the first study examining the mediating role of DNA methylation between prenatal anxiety and child behavioral problems.

## 5. Conclusions

We provided data supporting an association between maternal anxiety during pregnancy and DNA methylation changes of candidate regions in buccal cells of four year olds. Our findings add evidence to the moderating role of sex on the association of prenatal maternal anxiety with children’s epigenetic profile. Results were only found in the third trimester of pregnancy, indicating that advanced gestation may be a specifically vulnerable period for epigenetic modification by maternal anxiety. Additionally, we found several effects of sex of the child on changes of DNA methylation patterns in response to maternal anxiety during pregnancy, underscoring the importance of sex of the child for epigenetic processes early in life. Only weak evidence was found for a mechanistic role of child DNA methylation on the association between prenatal exposure to maternal anxiety and child behavioral measures at four years of age. Future gene expression studies should confirm whether DNA methylation changes correspond with altered gene expression in children exposed to maternal anxiety during pregnancy. The search for mechanisms to explain the lasting effects of maternal distress during pregnancy on offspring’s behavioral measures should therefore be continued and extended.

## Figures and Tables

**Figure 1 cells-10-02421-f001:**
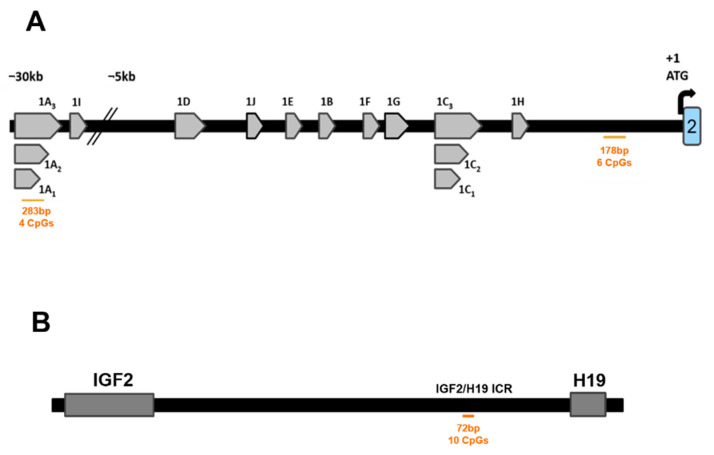
Structure of the human *NR3C1* and *IGF2/H19 ICR*. (**A**) Structure of the distal and proximal promoters of the human *NR3C1*. Numbers (in orange) represent the sequence fragments that were amplified in this study. (**B**) Structure of the *IGF2/H19 ICR*. Numbers (in orange) represent the sequence fragments that were amplified in this study.

**Figure 2 cells-10-02421-f002:**
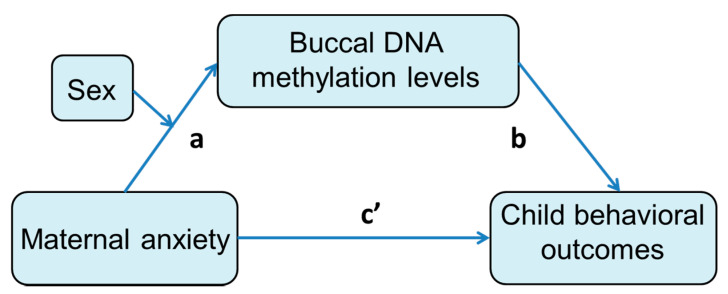
Theoretical model of the mediation analysis. Path a is the effect of the predictor variable on the DNA methylation (mediator), path b is the effect of the DNA methylation on the outcome variable controlling for the predictor variable, path c’ is the direct effect of the predictor variable on the outcome variable controlling for DNA methylation (mediator). The coefficient a × b represents the mediating effect of the predictor variable on the outcome variable through DNA methylation (mediator). For the moderated mediations, the index of the moderated mediation effect was (Maternal anxiety × Sex on Methylation) × (Methylation on outcomes) which tests whether the mediation is significantly moderated.

**Figure 3 cells-10-02421-f003:**
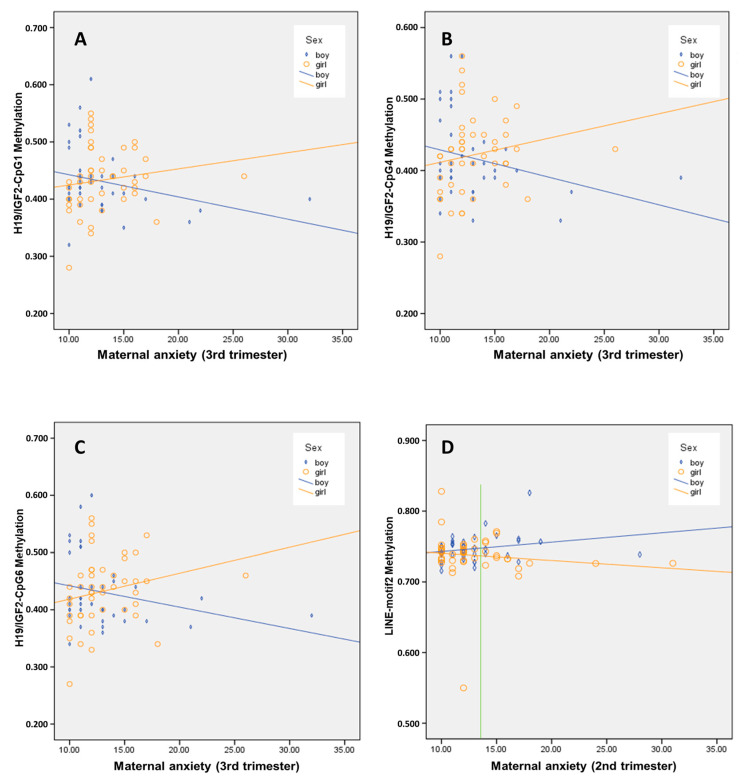
Scatter plots represent sex interaction. Blue squares indicate male and orange circles indicates female. Dashed blue line represents the fitting line in males and orange in females. X-axis represents the maternal anxiety score and Y-axis represents the beta-value of DNA methylation. (**A**–**C**) For boys, there was a negative association between maternal anxiety during the third trimester and the methylation level of *IGF2/H19* CpGs; for girls, there was a positive association between maternal anxiety during the third trimester and the methylation level of *IGF2/H19* CpGs. (**D**) For second trimester maternal anxiety level of 13.42 or higher there was a significant difference in DNA methylation level of *LINE1* motif 2 between boys and girls (*p* < 0.05). Green line indicates the value of 13.42.

**Figure 4 cells-10-02421-f004:**
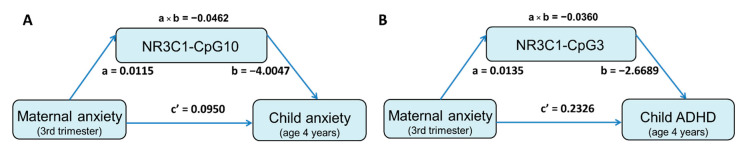
Mediation analyses on the association between exposure to prenatal maternal stress and child Behavioral measures. (**A**) The methylation level of NR3C1-CpG10 negatively mediated the effects of maternal anxiety during the third trimester on children’s anxiety (95% CI = (−0.1306) − (−0.0006)). (**B**) The methylation level of NR3C1-CpG3 negatively mediated the effects of maternal anxiety during the third trimester on children’s ADHD (95% CI = (−0.1190) − (−0.0019)).

**Table 1 cells-10-02421-t001:** Descriptive statistics of the mothers and children.

	**Range**	**Mean**	**Std. Deviation**
**Mothers**			
Gestational age at birth (weeks)	36.570–42.290	39.613	1.154
Anxiety—Trimester 1 (SCL-90)	10.000–35.000	13.794	4.741
Anxiety—Trimester 2 (SCL-90)	10.000–31.000	13.072	3.721
Anxiety—Trimester 3 (SCL-90)	10.000–32.000	13.072	3.581
Anxiety when child was at age of 4 (SCL-90)	10.000–32.000	12.771	4.058
	%		
**Children**			
Sex			
Male	44.900		
Female	55.100		
	**Range**	**Mean**	**Std. Deviation**
Children’s birth weight (g)	2605.000–4590.000	3427.360	408.427
Children’s behavior outcomes:			
-Internalizing problems (CBCL)	0.000–26.000	7.256	6.012
-Externalizing problems (CBCL)	1.000–34.000	10.683	6.887
-DSM affective problems (CBCL)	0.000–10.000	1.841	1.815
-DSM anxiety problems (CBCL)	0.000–8.000	2.268	1.969
-DSM pervasive developmental problems (CBCL)	0.000–14.000	3.378	2.765
-DSM attention deficit/hyperactivity problems (CBCL)	0.000–9.000	3.494	2.281
-DSM oppositional defiant problems (CBCL)	0.000–12.000	3.866	2.562

Notes. SCL-90 = Symptom Check List – 90; CBCL = Child Behavioral Checklist; DSM = Diagnostic and Statistical Manual of Mental Disorders; The cutoff for the SCL-90 anxiety subscale for “above average/high” for anxiety is 15 [55].

**Table 2 cells-10-02421-t002:** Correlation of children’s behavioral measures with maternal anxiety during pregnancy and with DNA methylation levels.

	CBCL Subscales						
	Internalizing Problems	Externalizing Problems	DSM Affective Problems	DSM Anxiety Problems	DSM Pervasive Developmental Problems	DSM Attention Deficit/Hyperactivity Problems	DSM Oppositional Defiant Problems
Anxiety—Trimester 1 (SCL-90)	0.052	0.053	0.108	−0.009	0.026	0.037	0.121
Anxiety—Trimester 2 (SCL-90)	0.254 ^a^	0.195	0.173	0.197	0.203	0.176	0.245 ^b^
Anxiety—Trimester 3 (SCL-90)	0.154	0.205	0.037	0.071	0.161	0.252 ^c^	0.203
*LINE1*_motif1	−0.061	0.051	−0.026	−0.038	−0.059	−0.014	0.087
*LINE1*_motif2	−0.126	0.078	−0.003	−0.097	−0.139	0.116	0.021
*LINE1*_motif3	−0.013	0.103	0.039	−0.037	0.008	0.067	0.123
*NR3C1*-CpG1	−0.008	−0.165	−0.026	−0.082	−0.028	−0.144	−0.105
*NR3C1*-CpG2	−0.042	−0.175	0.045	−0.100	−0.111	−0.177	−0.092
*NR3C1*-CpG3	−0.017	−0.151	0.019	−0.057	−0.086	−0.149	−0.085
*NR3C1*-CpG4	−0.016	−0.179	−0.009	−0.045	−0.044	−0.193	−0.136
*NR3C1*-CpG5	−0.057	0.029	−0.070	−0.135	−0.053	−0.065	0.006
*NR3C1*-CpG6	−0.079	−0.038	−0.090	−0.024	−0.070	−0.040	−0.088
*NR3C1*-CpG7	−0.143	−0.015	−0.147	−0.174	−0.064	−0.019	−0.052
*NR3C1*-CpG8	−0.106	−0.051	0.012	0.033	−0.100	−0.032	−0.023
*NR3C1*-CpG9	−0.076	0.014	−0.005	−0.157	−0.067	−0.084	0.081
*NR3C1*-CpG10	0.059	0.196	0.005	−0.168	0.066	0.010	0.201
*IGF2/H19 ICR* -CpG1	−0.054	−0.055	−0.002	−0.094	−0.067	−0.137	−0.020
*IGF2/H19 ICR* -CpG2	−0.091	−0.037	−0.039	−0.129	−0.104	−0.119	−0.001
*IGF2/H19 ICR* -CpG3	−0.070	−0.041	−0.023	−0.124	−0.087	−0.105	0.005
*IGF2/H19 ICR* -CpG4	−0.057	−0.041	−0.041	−0.107	−0.072	−0.093	−0.007
*IGF2/H19 ICR* -CpG5	−0.090	−0.062	−0.046	−0.123	−0.092	−0.104	−0.021
*IGF2/H19 ICR* -CpG6	−0.072	−0.008	−0.014	−0.116	−0.106	−0.076	0.028
*IGF2/H19 ICR* -CpG7	−0.089	−0.044	−0.042	−0.147	−0.116	−0.123	0.009
*IGF2/H19 ICR* -CpG8	−0.073	−0.021	−0.006	−0.129	−0.103	−0.107	0.036
*IGF2/H19 ICR* -CpG9	−0.072	−0.037	−0.081	−0.155	−0.101	−0.079	−0.009
*IGF2/H19 ICR* -CpG10	−0.092	−0.059	−0.111	−0.160	−0.113	−0.128	−0.018

Note. Pearson collection coefficients were shown. Significant values were labeled with ^a^ (*p* = 0.025), ^b^ (*p* = 0.03), and ^c^ (*p* = 0.026). SCL-90 = Symptom Check List–90; CBCL = Child Behavioral Checklist; DSM = Diagnostic and Statistical Manual of Mental Disorders.

**Table 3 cells-10-02421-t003:** Linear regression model of methylation level of *IGF2/H19*-CpG1.

Predictor Variable	β	R	R^2^	ΔR^2^	F	ΔF
		0.157	0.025		0.946	
Gestational age at birth	0.001					
Postnatal maternal anxiety	0.001					
		0.175	0.031	0.006	0.783	0.470
Gestational age at birth	0.001					
Postnatal maternal anxiety	0.001					
Maternal anxiety (third trimester)	−0.002					
		0.176	0.031	0.000	0.582	0.013
Gestational age at birth	0.001					
Postnatal maternal anxiety	0.001					
Maternal anxiety (third trimester)	−0.002					
Sex	0.002					
		0.291	0.085	0.054	1.336	4.246 ^a^
Gestational age at birth	0.000					
Postnatal maternal anxiety	0.001					
Maternal anxiety (third trimester)	−0.007					
Sex	−0.119					
Maternal anxiety (third trimester) × sex	0.009					

Note. Significant value was labeled with ^a^ (*p* = 0.043). β: Beta coefficients; R: Correlation coefficient; R^2^: R squared; ΔR^2^: change in R squared; F: F-score; ΔF: change in F-score.

**Table 4 cells-10-02421-t004:** Linear regression model of methylation level of *IGF2/H19*-CpG4.

Predictor Variable	β	R	R^2^	ΔR^2^	F	ΔF
		0.133	0.018		0.680	
Gestational age at birth	0.001					
Postnatal maternal anxiety	0.001					
		0.148	0.022	0.004	0.551	0.307
Gestational age at birth	0.001					
Postnatal maternal anxiety	0.001					
Maternal anxiety (third trimester)	−0.001					
		0.152	0.023	0.001	0.430	0.087
Gestational age at birth	0.001					
Postnatal maternal anxiety	0.001					
Maternal anxiety (third trimester)	−0.001					
Sex	0.004					
		0.291	0.085	0.061	1.329	4.836 ^a^
Gestational age at birth	0.000					
Postnatal maternal anxiety	0.001					
Maternal anxiety (third trimester)	−0.007					
Sex	−0.125					
Maternal anxiety (third trimester) × sex	0.010					

Note. Significant value was labeled with ^a^ (*p* = 0.031). β: Beta coefficients; R: Correlation coefficient; R^2^: R squared; ΔR^2^: change in R squared; F: F-score; ΔF: change in F-score.

**Table 5 cells-10-02421-t005:** Linear regression model of methylation level of *IGF2/H19*-CpG6.

Predictor Variable	β	R	R^2^	ΔR^2^	F	ΔF
		0.169	0.029		1.101	
Gestational age at birth	0.001					
Postnatal maternal anxiety	0.000					
		0.169	0.029	0.000	0.725	0.003
Gestational age at birth	0.001					
Postnatal maternal anxiety	0.000					
Maternal anxiety (third trimester)	0.000					
		0.169	0.029	0.000	0.536	0.000
Gestational age at birth	0.001					
Postnatal maternal anxiety	0.000					
Maternal anxiety (third trimester)	0.000					
Sex	0.000					
		0.297	0.088	0.059	1.390	4.697 ^a^
Gestational age at birth	0.001					
Postnatal maternal anxiety	0.000					
Maternal anxiety (third trimester)	−0.005					
Sex	−0.134					
Maternal anxiety (third trimester) × sex	0.011					

Note. Significant value was labeled with ^a^ (*p* = 0.034). β: Beta coefficients; R: Correlation coefficient; R^2^: R squared; ΔR^2^: change in R squared; F: F-score; ΔF: change in F-score.

**Table 6 cells-10-02421-t006:** Linear regression model of methylation level of *LINE1*-motif2.

Predictor Variable	β	R	R^2^	ΔR^2^	F	ΔF
		0.137	0.019		0.724	
Gestational age at birth	0.000					
Postnatal maternal anxiety	0.000					
		0.149	0.022	0.003	0.567	0.268
Gestational age at birth	0.000					
Postnatal maternal anxiety	0.001					
Maternal anxiety (second trimester)	0.000					
		0.211	0.045	0.022	0.846	1.738
Gestational age at birth	0.000					
Postnatal maternal anxiety	0.001					
Maternal anxiety (second trimester)	0.000					
Sex	−0.006					
		0.379	0.143	0.099	2.445	8.424 ^a^
Gestational age at birth	0.000					
Postnatal maternal anxiety	0.000					
Maternal anxiety (second trimester)	0.003					
Sex	0.049					
Maternal anxiety (second trimester) × sex	−0.004					

Note. Significant value was labeled with ^a^ (*p* = 0.005). β: Beta coefficients; R: Correlation coefficient; R^2^: R squared; ΔR^2^: change in R squared; F: F-score; ΔF: change in F-score.

## Data Availability

Data is stored at Tilburg University and is available on request by M.I.v.d.H.

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
