# Peer review of "Epigenetic Modifications Associated with Maternal Anxiety during Pregnancy and Children’s Behavioral Measures"

_cells, 2021, doi:10.3390/cells10092421_

Round 1
Reviewer 1 Report
This manuscript examines the influence of maternal anxiety during pregnancy on buccal cell DNA methylation and behavioural measurements in 4-year-old offspring. The authors focused on candidate genes/regions previously implicated in mediating the effects of prenatal stress on offspring neurodevelopment and behaviour (specifically NR3C1 and IGF2/H19). The strength of this study is its prospective design and evaluation of maternal anxiety in all three trimesters of pregnancy. The major limitation is the relatively small sample size. I find the idea behind the study interesting and the results worth publishing to contribute data that may prove useful in further exploring this hypothesis.
The analyses seem to have been done carefully in terms of statistical analyses and the manuscript is overall very well written, but there are some problems in the methods and data presentation that need to be addressed.
1) Number of participants: As stated in the manuscript (lines 151-155, 161-168) , 190 mothers (191 infants) were recruited and 107 were excluded. Clarification is needed as to why the final sample consisted of 89 (instead of 84) mother-infant dyads. In addition, according to Supplementary Table S2 anxiety measures in pregnancy were only available for 82/83 mothers, and since maternal anxiety in pregnancy was the main predictor, I believe dyads without these data should also be excluded.
2) It should be clarified whether child behavior was assessed directly or only by mothers who reported behavioral problems, and if the latter, this should also be mentioned as a study limitation. Was the Child Behavioral Check-List (CBLC) administered by psychologists/psychiatrists? The reference given for the CBCL (Ref 56) appears to be incorrect.
3) It should be reported whether the children had any diagnosis or were all healthy. I do not think I would refer to "DSM attention deficit/hyperactivity problems", "ADHD symptoms," or generally "behavior problems" in children who do not have ADHD or any other bihevioral disorder diagnosis. I would perhaps refer to these as "ADHD traits", “internalizing behavior”, “externalizing behavior”, “anxiety measures” etc. Also, I would suggest using "behavioral measures" instead of "behavioral problems" throughout the manuscript.
4) When buccal cells were collected, was there a standardized procedure prior to sampling? Please indicate the average yield and purity of DNA obtained from buccal cell samples.
5) Statistical analysis section: it is not clear in what way the distribution of the data was tested. The test used for correlation analyzes should be stated. Was there any procedure for outliers handling?
6) Please provide a range in Table 1 and include(in a supplement) descriptive statistics for children's characteristics (birth weight, gestational age, and behavioral measures) in boys and girls separately. Were these different between boys and girls? This is important for better interpretation of the moderating effect of sex on the associations studied.
7) For each of the CpG sites studied, the genomic position and descriptive statistics for methylation levels in the investigated sample should be provided. Were the methylation levels at nearby loci correlated? If so, perhaps the mean methylation level across the region would be a more reliable measure and would eliminate statistical problems with multiple testing.
8) Tables 2, 3, 4, 5, 6: Exact p-values should also be reported. Table 2 should indicate whether Pearson's or Spearmen's correlation coefficients are shown.
9) Figure 3: Please show data as a scatter plot instead of just lines, and report correlation coefficient and p-value in the graphs instead of the formulas that are currently shown. It is not clear what the values on the y-axis are - if it is the percentage of methylation, they seem to be extremely low (<1%); if it is something else, it needs to be explained.
10) Discussion:
line 417: it is not clear why CpG4 an 6 in IGF2/H19 are not mentioned
lines 431 and 506: the effect was not “strongest” in the 3rd trimester, but present only in the 3rd trimester of pregnancy
line 488: why “next to birth weight” – perhaps “next to gestational age”?
11) The title of the manuscript “Role of Epigenetic Modifications Involved in Maternal Anxiety During Pregnancy and Children’s Behavioral Problems “ is not clear, perhaps change it to “Epigenetic Modifications Associated with Maternal Anxiety During Pregnancy and Children’s Behavioral Measures”
Reviewer 2 Report
In this paper, Cao-Lei, van den Heuvel, et al. report that there are some links between maternal anxiety during pregnancy and children’s behavioral problems, and attempting to explain this correlation with DNA methylation status of specific genes. The reviewer acknowledges the importance of these type of cohort studies. Although the manuscript is descriptive, I also understand the technical difficulties to conduct these types of studies due to the sample limitations. I have couple of comments prior to acceptance.
Major comments
- The authors did not mention the ethnic background of the participants in the text. I would recommend including a brief description for this point.
- Did the DNA methylation changes observe in NR3C1 and IGF2/H19 loci correlate with actual gene expression? Can the authors provide any evidence that DNA methylation changes observed in this study have significant impact on gene expression? For instance, can the authors perform quantitative RT-PCR experiments for NR3C1 and IGF2/H19 genes? Just few samples should be enough.
- Related to previous comment but, how can the authors conclude that the DNA methylation changes observed are significant enough to explain the outcomes? In other words, how did the authors conclude that the DNA methylation changes observed between specimens are meaningful, without testing the gene expression?
- How can a maternal anxiety impact global DNA methylation level as observe by LINE1? Please discuss more thoroughly.
- Can the authors provide graph(s) for some of the Tables? It takes time to understand the meaning.
Round 2
Reviewer 1 Report
Please pay attention to the number of significant digits in data shown in Table 1 and the supplementary tables.
Tables 2 to 6: Please mark significant results with bold to make them easier to notice.
Author Response
We thank both reviewers for their extensive and helpful review. Please find a point-by-point response to the individual Reviewer’s comments, with the original Reviewer’s comments in italic and our response in normal typeset. Changes to the manuscript text are marked in Red in the manuscript file.
>>>Response: We have done the spell check.
Please pay attention to the number of significant digits in data shown in Table 1 and the supplementary tables.
>>>Response: We have modified the tables and keep the number of significant digits consistent.
Tables 2 to 6: Please mark significant results with bold to make them easier to notice.
>>>Response: We have marked significant results with bold.

Reviewer 2 Report
The authors gave satisfactory answers my questions. I have no further comments.
